# Intra-Cardiac versus Transesophageal Echocardiographic Guidance for Left Atrial Appendage Occlusion with a Watchman FLX Device

**DOI:** 10.3390/jcm12206658

**Published:** 2023-10-20

**Authors:** Luigi Emilio Pastormerlo, Claudio Tondo, Gaetano Fassini, Antonino Nicosia, Federico Ronco, Marco Contarini, Giuseppe Giacchi, Carmelo Grasso, Gavino Casu, Maria Rita Romeo, Patrizio Mazzone, Luca Limite, Giuseppe Caramanno, Salvatore Geraci, Paolo Pagnotta, Mauro Chiarito, Corrado Tamburino, Sergio Berti

**Affiliations:** 1Diagnostic and Interventional Cardiology Department, Fondazione Toscana Gabriele Monasterio, 54100 Massa, Italy; luigi.emilio.pastormerlo@ftgm.it (L.E.P.); mromeo@ftgm.it (M.R.R.); 2Department of Clinical Electrophysiology and Cardiac Pacing, Heart Rhythm Center at Monzino Cardiac Center, IRCCS, 20138 Milan, Italy; claudio.tondo@ccfm.it (C.T.); gfassini@ccfm.it (G.F.); 3Dipartimento Cardio-Neuro-Vascolare, Ospedale GP II—Asp di Ragusa, 97100 Ragusa, Italy; antonino.nicosia@asp.rg.it; 4Ospedale dell’Angelo di Mestre, 30174 Venice, Italy; federico.ronco@aulss3.veneto.it; 5Cardiology Department, Umberto I Hospital, ASP 8 Siracusa, 96100 Syracuse, Italy; marcocontarini@gmail.com (M.C.); peppegiacchi@gmail.com (G.G.); 6AOU Policlinico ‘G. Rodolico-San Marco’, Centro Alte Specialità e Trapianti—C.A.S.T., 95123 Catania, Italy; melfat75@gmail.com (C.G.); tambucor@unict.it (C.T.); 7Cardiologia Clinica e Interventistica, Azienda Ospedaliero Universitaria Sassari, 07100 Sassari, Italy; gcasu61@gmail.com; 8Department of Cardiac Electrophysiology and Arrhythmology, IRCCS San Raffaele Hospital, Vita-Salute University, 20132 Milan, Italy; mazzone.patrizio@hsr.it (P.M.); limite.luca@hsr.it (L.L.); 9Ospedale San Giovanni di Dio, 92100 Agrigento, Italy; giuseppe.caramanno@gmail.com (G.C.); totogeraci@yahoo.it (S.G.); 10Department of Biomedical Sciences, Humanitas University, 20072 Pieve Emanuele, Italy; paolo.pagnotta@humanitas.it (P.P.); chiaritomauro@gmail.com (M.C.); 11Humanitas Research Hospital IRCCS, 20089 Rozzano, Italy

**Keywords:** intra-cardiac, transesophageal, echocardiography, left atrial appendage occlusion

## Abstract

This study aimed to compare the peri-procedural success and complication rate within a large registry of intra-cardiac echocardiography (ICE)- vs. transesophageal echocardiography (TEE)-guided left atrial appendage occlusion (LAAO) procedures with a Watchmann FLX device. Data from 772 LAAO procedures, performed at 26 Italian centers, were reviewed. Technical success was considered as the final implant of a Watchmann FLX device in LAA; the absence of pericardial tamponade, peri-procedural stroke and/or systemic embolism, major bleeding and device embolization during the procedure was defined as a procedural success. One-year stroke and major bleeding rates were evaluated as outcome. ICE-guided LAA occlusion was performed in 149 patients, while TEE was used in 623 patients. Baseline characteristics were similar between the ICE and TEE groups. The technical success was 100% in both groups. Procedural success was also extremely high (98.5%), and was comparable between ICE (98.7%) and TEE (98.5%). ICE was associated with a slightly longer procedural time (73 *±* 31 vs. 61.9 *±* 36 min, *p* = 0.042) and shorter hospital stay (5.3 *±* 4 vs. 5.8 *±* 6 days, *p* = 0.028) compared to the TEE group. At one year, stroke and major bleeding rates did not differ between the ICE and TEE groups. A Watchmann FLX device showed high technical and procedural success rate, and ICE guidance does not appear inferior to TEE.

## 1. Introduction

Atrial fibrillation (AF) is the most common cardiac arrhythmia, affecting 3–5% of people older than 65 years [1,2]. Oral anticoagulation is mandatory in patients with AF in order to minimize the risk of thromboembolic events. Nevertheless, nearly 40% of patients at risk of stroke do not receive any form of oral anticoagulation in the real world [3,4,5]. Percutaneous left atrial appendage occlusion (LAAO) has been demonstrated as a viable alternative for thromboembolic event prevention in patients with absolute/relative contraindications to (novel) oral anticoagulants (NOACs) [6]. The procedural safety of LAAO is a key element, as it may affect the procedural results and patient follow-up. It is also crucial in order to propose a similar procedure as an alternative to NOACs also in patients that can tolerate NOACs. Many elements may affect procedural safety. Significant advances have been registered within intra-procedural imaging and technological improvements of the dedicated devices.

In last years, new devices or new versions of past devices have emerged. Watchman FLX represents an upgrade of the Watchman family devices. In comparison to the old version of the device, Watchman FLX has greater flexibility and reduced height, with a higher number of anchoring elements, improved fabric coverage and a closed distal end. All these modifications are aimed for an easier implantation, safer recapture and repositioning, with the potential for reduction in the follow-up of device-related thrombosis and peri-device leaks. The high flexibility of the device permits for a significant oversize (in order to limit the device embolization risk), without increasing the risk of perforation. Watchmann FLX has demonstrated optimal safety and efficacy results, with a low rate of procedural complications.

On the other hand, intra-procedural imaging may significantly affect procedural outcome. The optimal way to guide an LAAO procedure has not been completely defined. The use of fluoroscopy without other imaging modality should be strongly discouraged, as it is inadequate for a detailed guidance of an LAAO procedure. Transesophageal echocardiography (TEE) has been the most common imaging modality adopted for LAAO guidance. It is well known by many cardiologists in the context of standardized imaging and optimal visualization of the LAA, atrial septum, left and right atria. Nevertheless, the needed general anesthesia and dedicated operator are two main limits of TEE use. Moreover, interference with fluoroscopy, as well as the potential for upper gastrointestinal tract damage, is another factor that may affect TEE use. Intra-cardiac echocardiography (ICE) has been developed as an alternative to TEE for LAAO procedure guidance. ICE probes are commonly advanced through the right femoral vein, and positioned close to the structure of interest for an LAAO in the right and, after transeptal puncture, in the left atria. 

Available data suggest that ICE is not inferior to TEE for guiding LAAO procedures, both for immediate results (procedural success and peri-procedural complications) and long-term patient outcomes [7,8,9]. A large registry has specifically demonstrated that the Amulet device implantation under ICE guidance is not inferior to TEE in terms of procedural safety and patient outcomes [10].

In the present work, we aim to compare ICE to TEE guidance for an LAAO procedure within a large Italian registry of LAAO procedures performed with the Watchmann FLX device. 

## 2. Methods

### 2.1. Study Population

We prospectively enrolled 772 patients that underwent LAA occlusion procedures at 26 Italian centers, between October 2018 and September 2021. Procedural and follow-up data were collected. Patients with non-valvular atrial fibrillation, a CHA2DS2-VASc score ≥ 2 and relative/absolute contra-indication for NOACs were included. The CHA_2_DS_2_-VASc [11] and HAS-BLED [12] scores were calculated. Informed consent was collected from each patient. The local ethics Committee “(Comitato Etico Area Vasta Nord Ovest)” approved this study. 

### 2.2. Device and Implantation Procedure

All procedures were performed with the Watchmann FLX device (Boston Scientific, Marlborough, MA, USA). The device was recapturable and repositionable at any moment. A higher number of open-frame cells increases device conformability, while a closed end facilitates device implantation when the length of the LAA is limited (minimum required length is 50% of the device size). The risk of perforation is limited by the presence of an atraumatic distal end with a fluoroscopic marker. This design permits for device advancement within the LAA when partially exposed in a “ball” configuration. Still, the polyester fabric coverage has been extended to minimize peri-device leaks. The device is available in five sizes, namely 20, 24, 27, 31 and 35 mm for LAA ostia sizes ranging from 14 mm to 31.5 mm.

Preprocedural evaluation and planning was performed with transesophageal echocardiography (TEE) or cardiac computed tomography (CT), according to the local practice. LAA dimensions and morphology were evaluated, as well as the morphology of intertribal septum and fossa ovalis. The landing zone (LZ) was measured at the LAA ostium, from the level of the circumflex coronary artery to a point 1-to-2 cm distal to the left upper pulmonary vein ridge. Three-dimensional reconstruction was used. The device sizing was selected according to the maximal LAA LZ diameter, with target oversizing between 10% and 30%. Patients with LAA thrombosis were excluded. 

TEE or ICE were used during the procedure for guidance of the different phases, namely transseptal puncture, LAA morphology and size reevaluation, LAA incanulation, device delivery and implantation (after an assessment of the position and stability, leak absence). Procedural complications were promptly evaluated. TEE procedures required general anesthesia. ICE procedures required an adjunctive ipsilateral femoral vein access and were performed under local anesthesia. An alternative use of ICE or TEE was left to the operator discretion and TEE/ICE availability.

For optimal ICE guidance, an ICE probe was advanced into the left atrium during every ICE-guided procedure, through the first transseptal puncture, as described elsewhere.

Technical success was defined as the final deployment of the device inside an LAA, fully meeting the PASS criteria (position: device at the LAA ostium; anchor: fixation anchors engaged/stable device; size: 8 to 20% device compression; seal: device spans the ostium, all lobes covered). Procedural success was defined as technical success without major procedure-related complications (pericardial tamponade requiring drainage, stroke, systemic embolism, major bleeding and device embolization requiring surgical removal, absence of a final leak > 5 mm). Major bleedings were defined according to the International Society on Thrombosis and Haemostasis criteria [12]. Events were labeled as peri-procedural if they occurred within 7 days of the procedure or pre-discharge.

### 2.3. Follow-Up

Follow-up was performed by clinical visits or telephonic follow-up. No oral anticoagulation therapy was administered after the procedure. Different pharmacological schemes were adopted after the procedure. The most common scheme included dual antiplatelet therapy with aspirin (100 mg/die) plus clopidogrel (75 mg/24 h) for 30 to 180 days after the procedure. According to operators’ discretion, in patients with high bleeding risk, a single antiplatelet agent was prescribed with/without adjunction of a prophylactic dose of low-molecular-weight heparin for the first month after the implant procedure. 

Stroke, transitory ischemic attacks (TIA), systemic embolism, major/minor hemorrhage and death from cardiac or any cause were considered as events. Patients that had a one year follow-up were considered for the outcome analysis. 

### 2.4. Statistical Analysis

Continuous data were expressed as the mean ± standard deviation. The procedural success rate was calculated as a percentage over the total number of procedures. For proportions, numbers and percentages were used. The differences between TEE and ICE groups were evaluated with the Student t-test for continuous variables. Binary logistic regression analysis was used to evaluate the occurrence of complications in the TEE and ICE groups, respectively. The odds ratios and 95% confidence intervals were calculated. Events curves were analyzed using the Kaplan–Meier estimate. The differences in survival curves were tested with a log-rank test. Statistical analysis was performed using SPSS^®^ (version 19, SPSS, Chicago, IL, USA). A *p*-value less than 0.05 was considered significant.

## 3. Results

### 3.1. Patient Population

The baseline characteristics, as well indications for LAAO, are displayed in Table 1 (overall population, TEE and ICE groups, respectively). 

Of the total of 772 procedures, ICE-guided LAA occlusion was performed in 149 patients, while TEE was used in 623 patients. Clinical characteristics and indications for the procedure were similar between the groups, but patients in the ICE group had a higher prevalence of hematologic disorders. About 50% of the patients had a permanent AF, without any difference between TEE and ICE patients. CHA_2_DS_2_-VASc was 4.1 and HAS-BLED 3.6, with no differences between TEE and ICE patients. Of patients, 91% in the TEE group and 8% of ICE patients underwent pre-procedural three-dimensional TEE, while pre-procedural CCT was performed in 10% of the patients in the TEE group and 94% of ICE patients, respectively.

### 3.2. Procedural Results and In-Hospital Outcomes

Table 2 shows procedural characteristics, complication rate and in-hospital outcomes across the entire population and within the TEE and ICE groups, respectively. 

The use of ICE was associated with longer procedural (73 ± 31 vs. 61.9 ± 36 min, *p* = 0.042) and fluoroscopy times (24 ± 15 vs. 17 ± 13 min, *p* = 0.031) in comparison to the ICE group. On the other side ICE patients had shorter hospital stay (5.3 ± 4 vs. 5.8 ± 6 days, *p* = 0.028). No differences in the number of attempts for device positioning, as well as the number of device opened, were assessed between the ICE and TEE groups. The overall technical success was 100%, with the final fully meeting the PASS criteria for all procedures. Procedural success was similarly high, with no differences between the TEE (98.5%) and ICE (98.7%) groups and a complication rate of 1.5% for the TEE vs. 1.3% for the ICE, respectively. The most common complication was major pericardial effusion for both groups, followed by major bleeding. Of the pericardial effusions, 25% required drainage, 0.55% in the ICE group and 0.47% in the TEE group. Any case of device embolization was reported, as well as any case of peri-device leak > 5 mm that occurred. Procedural complications in the ICE- vs. TEE-guided procedure groups are visualized in Figure 1.

### 3.3. Follow-Up

One year follow-up data were available for 342 patients, with 102 in the ICE group. The rate of stroke (1.9% in the ICE group vs. 2% in the TEE group, *p* = ns) and major bleeding (2.9% in the ICE group vs. 3.7% in the TEE group) were similar between the ICE and TEE groups (Figure 2, Table 3).

## 4. Discussion

This large, multicenter, Italian registry on LAA occlusion with a Watchman FLX device has two main messages: (a) a Watchman FLX device ensures high procedural safety and technical success; (b) ICE intra-procedural guidance for LAA occlusion does not appear to be inferior to TEE in terms of procedural complications and one year follow-up outcomes.

In the last 20 years, LAAO has been demonstrated as a viable alternative to (N)OACS in NVAF patients with absolute/relative contraindications to anticoagulation. Specifically, LAA occlusion reduces about 70% of the rate of cerebral embolism as compared to historical NVAF populations without OAC, at the same time warranting a 60% reduction in the rate of hemorrhagic events as compared to historical populations under OAC treatment [13].

In this context, procedural safety is a key element. LAAO history has been characterized by progressive amelioration of the procedural safety profile, making LAAO more attractive. A low rate of procedural complications is crucial in order to implement LAAO in current clinical practice as a potential alternative to NOACs, also in patients who could tolerate this therapy.

Two main procedural aspects need full consideration regarding procedural safety. The first element is about technical equipment and, specifically, the device. Different devices have been developed. The Watchmann family devices have been the most studied devices both, in randomized trials [14] and registries [6]. Watchman FLX is the last version of this family of devices, which has been developed with the intent to optimize procedural results in comparison to its previous version and other devices. Thanks to its improved flexibility and reduced height, this device has a wide range of size adaptations, allowing for a high grade of oversize without affecting the procedural safety. This characteristic, along with the presence of an additional row of 18 J-shaped anchors, has the advantage of minimizing the risk of device embolization, with any case of device embolization in the present registry. Still, the flexibility and closed distal end simplify implantation, allowing for full and safe recapture and repositioning. The device full coverage, finally, is intended to reduce peri-device leaks and device-related thrombosis.

The second main aspect for improving procedural safety lays on accurate pre- and intra-procedural imaging. Pre-procedural-computed tomography or 3D high-quality TEE are essential. They inform about interatrial septum anatomy and its spatial relationship with the LAA (in order to optimize transseptal puncture), and must give full elucidation of LAA morphology. LAA has a wide variety of anatomical aspects, width of its neck, length, orientation and the presence of additional lobes must be fully analyzed before the procedure.

Intra-procedural imaging guides the procedure, from transeptal puncture to the device implantation and final assessment. TEE has historically been considered the gold standard for intra-procedural guidance. TEE certainly has a high definition of the right and left atria, atrial septum and left atrial appendage anatomy, and it is an imaging modality that is well known by many invasive cardiologists with largely standardized imaging. Nevertheless TEE may be affected by some limitations. First, TEE requires general anesthesia with a longer patient turnover and increased procedural time, implying the presence of an anesthesiologist. Still, general anesthesia may be harmful in fragile patients and may require intensive care monitoring after the procedure. Finally, TEE and endotracheal intubation may contribute to complications such as the injuries and bleeding of gastroesophageal and respiratory tract, laryngospasm and bronchospasm. requiring additional intervention and prolonged hospitalization [15]. ICE has been proposed as an alternative to TEE for LAA occlusion procedure guidance, and its use has been validated in large cohorts of patients. It has been demonstrated that even if ICE may result in an inferior image quality, this does not affect the procedural success and patient outcomes. What is crucial to implement for ICE use, along with the quality of pre-procedural evaluation, is placement of the ICE probe in left atrium (through the same transseptal puncture performed for device positioning). With the ICE probe in the left atrium, imaging quality is not inferior or even superior to TEE [16].

In this registry, ICE did not appear inferior to TEE in terms of procedural safety and follow-up results, using a very-well-performing device. The procedural success appears to be extremely high. In particular, the absence of device embolization and peri-device major leaks attest the high performance of this device, not affected by the choice of ICE or TEE guidance. TEE has lower fluoroscopy time, procedural time and contrast volume vs. ICE in this registry. These differences are likely linked to a higher standardization and operators experience with TEE guidance. A longer procedural time may be reduced in the future by higher operator experience, and, in any case, is counterbalanced by a lower cathlab turnover time and less invasiveness than ICE.

Still, one year thromboembolic and hemorrhagic event rates are extremely low, both in the ICE and TEE groups, if related to the CHA_2_DS_2_-VASc and HAS-BLED scores of the studied population.

These considerations pave the way for broader indications for an LAAO procedure. First, randomized trials are ongoing, evaluating the comparison between novel anticoagulants and LAAO. The results are awaited in the next months, and it is likely that many patients could also tolerate oral anticoagulant therapy, a potential candidate for LAAO. In this context, there is potential for a larger use of a combined procedure of LAAO and AF ablation. Patients that undergo AF ablation are also natural candidates to LAAO if this procedure shows good safety and efficacy [17].

As for many other fields of clinical practice, it is likely that artificial intelligence will impact on the indications and the way cardiac structural intervention performance. Some experience of pre-procedural planning and procedure simulation driven by artificial intelligence has been already published for LAAO [18,19]. In the future, artificial intelligence systems will have a key role in every aspect of patient care, from accurate patients selection, suggestion of best device and best imaging modality choice, with fine and accurate prediction of procedural results and patient outcomes.

## 5. Study Limitations

The main limitation of this study is the lack of randomization between ICE and TEE use. In this series, ICE was used mainly at centers with a high volume of LAA occlusion procedures. Follow-up data were available only for patients that achieved 1 year follow-up, which is actually about half of the study population. The study is largely underpowered to draw a definite conclusion about the follow-up.

## 6. Conclusions

A Watchmann FLX device has very high technical and procedural success rates. ICE guidance does not appear inferior to TEE in terms of procedural results and outcomes.

## Figures and Tables

**Figure 1 jcm-12-06658-f001:**
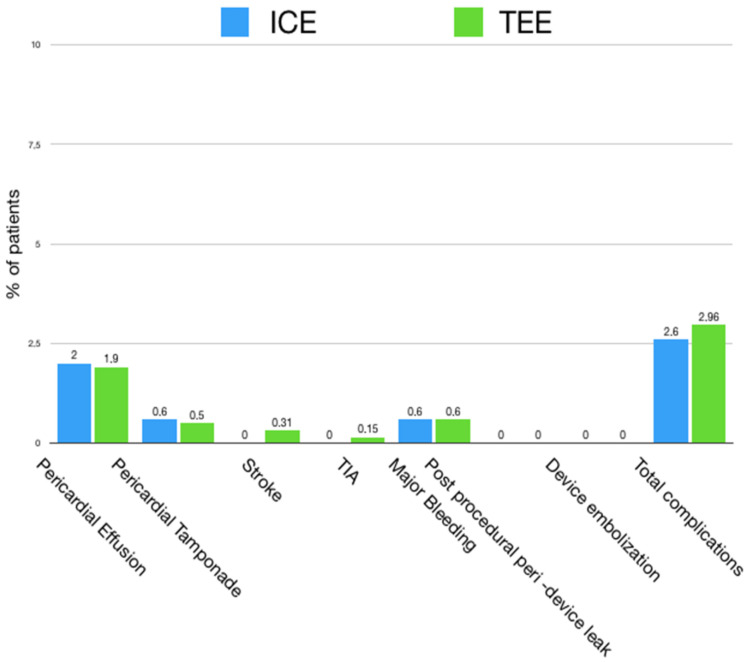
Procedural complications for ICE- (light blue) vs. TEE (green)-guided LAAO procedures. ICE: intra-cardiac echocardiography; LAAO: left atrial appendage occlusion; TEE: transesophageal echocardiography, TIA: transient ischemic attack.

**Figure 2 jcm-12-06658-f002:**
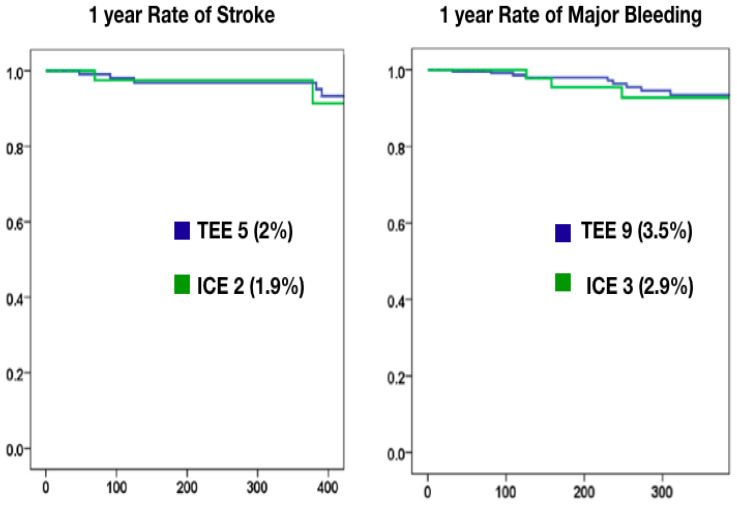
Kaplan–Meier curves at one year for stroke (**left**) and major bleeding (**right**) for ICE- vs. TEE-guided LAAO procedures. ICE: intra-cardiac echocardiography; LAAO: left atrial appendage occlusion; TEE: transesophageal echocardiography.

**Table 1 jcm-12-06658-t001:** Clinical details of patients undergoing an ICE- or TEE-guided LAAO procedure.

Patient	ALL (n = 772)	ICE (n = 149)	TEE (623)	*p*
Age	76.5 ± 8.3	77 ± 7.5	76.3 ± 8	ns
Male, %	509 (65%)	97 (65%)	407 (65%)	ns
CHA_2_DS_2_VASC	4.14 ± 1.47	4.2 ± 1.8	4.1 ± 1.4	ns
HASBLED	3.69 ± 1.1	3.5 ± 1.4	3.7 ± 1.1	ns
Creatinine	1.4 ± 1	1.3 ± 1.1	1.44 ± 1.2	ns
eGFR	59 ± 30	60 ± 28	58.4 ± 28	ns
LVEF	52 ± 10	54 ± 11	51 ± 11	ns
BMI	26.5 ± 3	26 ± 4	26.7 ± 4.4	ns
Permanent AF, %	376 (48%)	72 (48%)	304 (48%)	ns
Diabetes	264 (33%)	45 (30%)	219 (34%)	ns
Arterial hypertension	606 (78%)	115 (77%)	491 (78%)	ns
Previous stroke	106 (13%)	19 (12%)	87 (13%)	ns
GI bleeding	240 (30%)	40 (26%)	200 (31%)	ns
Intra-cranial/subdural bleeding	115 (14%)	25 (16%)	90 (14%)	ns
Severe CKD	126 (16%)	18 (12%)	108 (17%)	ns
Hematologic disorder	102 (13%)	28 (18%)	74 (11%)	0.038
Ischemic stroke in OACs	54 (7%)	8 (5.5%)	46 (7.3%)	ns
DAPT+OACs	7 (0.9%)	2 (1.3%)	5 (0.8%)	ns
Very high bleeding risk	122 (15%)	25 (16%)	97 (15%)	ns

AF: atrial fibrillation; BMI: body mass index; CKD: chromic kidney disease; DAPT: dual antiplatelet therpy; GFR: glomerular filtration rate; GI: gastrointestinal; ICE: intra-cardiac echocardiography; LAAO: left atrial appendage occlusion; LVEF: left ventricular ejection fraction; OACs: oral anticoagulants; TEE: transesophageal echocardiography.

**Table 2 jcm-12-06658-t002:** Procedural details of the TEE- or ICE-guided LAAO procedures, respectively.

	ALL	ICE	TEE	*p*
**Patients**	772	149	623	
**Pericardial effusion**	15 (1.9%)	3 (2%)	12 (1.9%)	NS
**Pericardial tamponade**	4 (0.5%)	1 (0.55%)	3 (0.47%)	NS
**Stroke**	2 (0.25%)	0 (0%)	2 (0.31%)	NS
**TIA**	1 (0.13%)	0 (0%)	1 (0.15%)	NS
**Device embolization**	0 (0%)	0 (0%)	0 (0%)	NS
**Peri-device leak > 5 mm**	0 (0%)	0 (0%)	0 (0%)	NS
**Major bleeding**	5 (0.6%)	1 (0.6%)	4 (0.63%)	NS
**Death**	0 (0%)	0 (0%)	0 (0%)	NS
**Technical success**	772 (100%)	149 (100%)	623 (100%)	NS
**Procedural success**	763 (98.5%)	147 (98.7%)	616 (98.5%)	NS
**Fluoroscopy time**	18.7 ± 14	24 ± 15	17 ± 13	0.031
**Procedure time**	63.8 ± 34	73 ± 31	61.9 ± 36	0.042
**Contrast volume**	92.11 ± 58	107 ± 53	88 ± 59	0.071
**In-hospital stay (days)**	5.6 ± 5	5.3 ± 4	5.8 ± 6	0.028

ICE: intra-cardiac echocardiography; LAAO: left atrial appendage occlusion; TIA: transitory ischemic attack; TEE: transesophageal echocardiography.

**Table 3 jcm-12-06658-t003:** One year follow-up results in patients that underwent an ICE- or TEE-guided LAAO procedure.

	ALL (n = 342)	ICE (n = 102)	TEE (n = 240)	*p*
Stroke	7 (2%)	2 (1.9%)	5 (2%)	NS
TIA	2 (0.5%)	0 (0%)	2 (0.8%)	NS
Death	11 (3.2%)	3 (2.9%)	8 (3.3%)	NS
Major bleeding	12 (3.5%)	3 (2.9%)	9 (3.7%)	NS

ICE: intracardiac echocardiography; LAAO: left atrial appendage occlusion; TIA: transitory ischemic attack; TEE: transesophageal echocardiography.

## Data Availability

Data set may be requested to pastormerlo@ftgm.it.

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
