# Peer review of "Intra-Cardiac versus Transesophageal Echocardiographic Guidance for Left Atrial Appendage Occlusion with a Watchman FLX Device"

_jcm, 2023, doi:10.3390/jcm12206658_

Round 1

Reviewer 1 Report

I have reviewed the manuscript entitled ‘Intra-Cardiac versus Transesophageal Echocardiographic Guidance for Left Atrial Appendage Occlusion with Watchman FLX device’. The manuscript aims to compare ICE and TEE in the interventions of LAAO which is very popular and has a potential to be a clinical routine in several countries. In the study patients, half of them have permanent AF, how about the other half of the patients? Are they in sinus rhythm or did they undergo AF ablation? The role of AF ablation in frail patients and patients with heart failure is very important. AF ablation can also reduce mortality in patients with heart failure. Please add a short section to the discussion and consider citing ‘Comparison of catheter ablation and medical therapy for atrial fibrillation in heart failure patients: A meta-analysis of randomized controlled trials’In the study there are no differences in terms of the intervention success. However, there are notable differences in fluoroscopy time, procedural time and contrast volume which are all in favour of TEE. This should be due to the experience with TEE and ICE. This difference should be mentioned in the discussion. The importance of LAAO will increase in a near future. We will include more patients for LAAO intervention since there are very high-risk patients for thromboembolism even under the treatment of NOACs.  The role of Artifical intelligence system will help us to determine perfoming this procedure to precise patients in a near future. Please add a short section to the discussion citing ‘The Role of Artificial Intelligence in Coronary Artery Disease and Atrial Fibrillation’

The language is fine.

Author Response

REVIEWER 1

We thank reviewer 1 for his/her constructive criticism.

I have reviewed the manuscript entitled ‘Intra-Cardiac versus Transesophageal Echocardiographic Guidance for Left Atrial Appendage Occlusion with Watchman FLX device’. The manuscript aims to compare ICE and TEE in the interventions of LAAO which is very popular and has a potential to be a clinical routine in several countries. 

In the study patients, half of them have permanent AF, how about the other half of the patients? Are they in sinus rhythm or did they undergo AF ablation? The role of AF ablation in frail patients and patients with heart failure is very important. AF ablation can also reduce mortality in patients with heart failure. Please add a short section to the discussion and consider citing ‘Comparison of catheter ablation and medical therapy for atrial fibrillation in heart failure patients: A meta-analysis of randomized controlled trials’

The percentage of patients that underwent AF ablation was very low in this patient group. We added the following paragraph in the discussion section to consider the role of AF ablation in patients that underwent LAAO, the suggested citation has been added: 
“First of all randomized trial are ongoing evaluating comparison between novel anticoagulants and LAAO;  Results are awaited in next months and it is likely that also many patients that could tolerate oral anticoagulant therapy will be potential candidate for LAAO. In this context there is potential for larger use of combined procedure of LAAO and AF ablation. Patients that undergo AF ablation are natural candidate also to LAAO if this procedure has safety and efficacy that approximate 100%. On the other side AF ablation is a  procedure that must be considered also in more frail patients and patients with heart failure given the demonstration of outcome advantage also in this patients subgroups [18]. “

In the study there are no differences in terms of the intervention success. However, there are notable differences in fluoroscopy time, procedural time and contrast volume which are all in favour of TEE. This should be due to the experience with TEE and ICE. This difference should be mentioned in the discussion. 

This difference is probably due to higher TEE standardization. We have mentioned this difference in the discussion. The following sentence has been added to the discussion 
“ TEE has lower fluoroscopy time, procedural time and contrast volume vs ICE in this registry. These differences are likely linked to higher standardization and operators experience with TEE guidance. Longer procedural time may be reduced in the future by higher operator experience and in any case is counterbalanced by lower cathlab turnover time and less invasiveness of ICE.  “  

The importance of LAAO will increase in a near future. We will include more patients for LAAO intervention since there are very high-risk patients for thromboembolism even under the treatment of NOACs.  The role of Artifical intelligence system will help us to determine perfoming this procedure to precise patients in a near future. Please add a short section to the discussion citing ‘The Role of Artificial Intelligence in Coronary Artery Disease and Atrial Fibrillation’

A brief paragraph about the role of artificial intelligence system has been added as suggested. The suggested citation has been added:
As for many other fields of clinical practice, it is likely that artificial intelligence will impact on indications and way of perform of cardiac structural interventions. Some experience of pre-procedural planning and procedure simulation driven by artificial intelligence have been already published for LAAO [19]. In the next future, artificial intelligence systems will have a key role in  every aspect of patient care, from accurate patients selection, suggestion of best device and best imaging modality choice, with fine and accurate prediction of procedural results and patient outcome.

Reviewer 2 Report

Well-designed and well-presented study based on a large and long-lasting Italian LAAO registry from experienced authors with ample publication background in this field. The study clearly presents the advantages of the latest version of Watchman device and the noninferiority of ICE (offering advantages compared with TEE - both for patient comfort and safety and for cathlab turnover) for guiding the procedure.

Author Response

REVIEWER 2 
Well-designed and well-presented study based on a large and long-lasting Italian LAAO registry from experienced authors with ample publication background in this field. The study clearly presents the advantages of the latest version of Watchman device and the noninferiority of ICE (offering advantages compared with TEE - both for patient comfort and safety and for cathlab turnover) for guiding the procedure.

We thank reviewer 2 for his/her appreciation.